# Transient Receptor Potential Channels and Itch

**DOI:** 10.3390/ijms24010420

**Published:** 2022-12-27

**Authors:** Omar Mahmoud, Georgia Biazus Soares, Gil Yosipovitch

**Affiliations:** Dr. Philip Frost Department of Dermatology and Cutaneous Surgery, Miami Itch Center, University of Miami Miller School of Medicine, Miami, FL 33136, USA

**Keywords:** transient receptor potential channel, TRP channel, TRPV, TRPA, TRPC, TRPM, itch, pruritus, agonist, antagonist

## Abstract

Transient Receptor Potential (TRP) channels are multifunctional sensory molecules that are abundant in the skin and are involved in the sensory pathways of itch, pain, and inflammation. In this review article, we explore the complex physiology of different TRP channels, their role in modulating itch sensation, and their contributions to the pathophysiology of acute and chronic itch conditions. We also cover small molecule and topical TRP channel agents that are emerging as potential anti-pruritic treatments; some of which have shown great promise, with a few treatments advancing into clinical trials—namely, TRPV1, TRPV3, TRPA1, and TRPM8 targets. Lastly, we touch on possible ethnic differences in TRP channel genetic polymorphisms and how this may affect treatment response to TRP channel targets. Further controlled studies on the safety and efficacy of these emerging treatments is needed before clinical use.

## 1. Introduction

Itch is defined as an unpleasant sensation that results in the urge to scratch. The sensation occurs when pruritogens activate receptors on cutaneous sensory C-fibers. There are two types of receptors that transmit itch: G protein-coupled receptors (GPCRs) and downstream transient receptor potential channels (TRP) [1]. TRP channels are a superfamily of cation permeable channels that are involved in a variety of sensory modalities [2]. The opening of TRP channels leads to depolarization of neuronal membranes and processing of pruriceptive signals, causing itch sensation [1]. Itch sensation can be acute or chronic (>6 weeks) and can be mediated by histaminergic and nonhistaminergic pathways, both of which require TRP signaling [Figure 1]. In this section, we will discuss the different types of TRP channels involved in pruritus, the role they play in pruritic diseases, and the potential antipruritic effects of compounds targeting these channels [Table 1].

## 2. TRP Ankyrin 1 (TRPA1)

TRPA1 plays an important role in histamine-independent itch, a process that underlies many chronic itch diseases [3]. The TRPA1 receptor has been found on keratinocytes, fibroblasts, melanocytes, dorsal root ganglia, and nerve c-fibers [4]. TRPA1 acts as a cold sensitive calcium channel in keratinocytes—in response to cold temperatures (less than 17 degrees Celsius), the channel is activated resulting in increased intracellular calcium [4,5]. The channel is also activated by various endogenous and exogenous pruritogens, many of which are noxious agents, which then act on downstream GPCRs or by directly stimulating TRPA1 channels [6]. Exogenous pruritogen activators of this channel include the anti-malarial drug chloroquine, cowhage, allyl isothiocyanate (an alkaloid found in substances such as mustard oil, wasabi, and horseradish), cinnamaldehyde (substance found in cinnamon oil), allicin (substance found in garlic), and carvacrol. Endogenous itch mediators that activate TRPA1 include leukotriene B4, thymic stromal lymphopoietin, serotonin, bile acids, and certain interleukins, namely IL-13 and IL-31 [5,7].

TRPA1 is thought to play a role in the common side effect of resistant itch associated with the anti-malarial drug chloroquine. This is mediated in part by specific Mas-related G protein-coupled receptors (Mrgpr) that can act on downstream TRPA1 channels to contribute to itch sensation. Chloroquine and the endogenous opioid peptide, bovine adrenal medulla 8–22 (BAM8-22), are pruritogens that activate MrgprA3 and MrgprC11, respectively, to induce itch [6]. Interestingly, TRPA1 knockout significantly reduced itching caused by chloroquine and BAM 8–22 in mice. MrgprX1 is a homologous channel in humans which suggests it may have a role in nonhistaminergic itch and may be a potential target for therapy [4]. However, a study investigating the role of MrgprX1 in itch sensation found that activation of MrgprX1 induces itch sensation through increased activity of tetrodotoxin-resistant voltage-gated sodium channel, not through TRP channels [8]. 

There are abundant animal model studies highlighting the importance and role of TRPA1 in the process of itch. TRPA1 was found to be important in generating spontaneous scratch in mouse models [9]. Additionally, when TRPA1 is knocked out, as demonstrated by atopic dermatitis mouse models, the mice were found to have a decreased scratching response [10]. A study by Nattkemper et al. biopsied itchy lesional and non-itchy lesional skin of patients with atopic dermatitis and found increased gene expression of TRPA1 in itchy skin of atopic dermatitis patients highly correlated to itch intensity [11]. TRPA1 also plays a role in itch associated with allergic contact dermatitis [7]. Genetically ablating or blocking TRPA1 pharmacologically alleviated itch in mice with oxazolone- and urushiol- induced dermatitis. Similarly, another study found impaired scratching response in TRPA1 and HTR7 knockout mice with vitamin D analog induced dermatitis [6,12]. 

There are many other pruritogenic pathways and mechanisms by which TRPA1 is associated with to contribute to itch. Increased interleukin-31 (IL-31) is found in many itchy conditions including atopic dermatitis and cutaneous T-cell lymphoma, and is thought to contribute to itch sensation through modulation of TRPA1 [13,14]. IL-31 has been shown to upregulate TRPA1 transcription and expression in itchy lesional skin in atopic dermatitis [15]. IL-31 also directly stimulates sensory neurons by binding to its receptor which opens TRPA1 (and TRPV1 channels) [16]. Itch secondary to IL-31 is decreased significantly when TRPA1 is knocked out—suggesting a possible mechanism of treating chronic itch [17]. Thymic stromal lymphopoietin (TSLP) is an important cytokine produced by epithelial cells and is also found to play a role in atopic dermatitis. TSLP expression is increased in the skin of patients with atopic dermatitis and in mouse models of atopic dermatitis [18]. TSLP can bind to its receptor directly on dorsal root ganglia of neurons, and through downstream signaling via phospholipase C, can induce pruritus with the help of TRPA1 [6,19]. This process is mediated by protease-activated receptor 2 (PAR2), which plays an important role in TSLP production in keratinocytes. TSLP has also been shown to increase TRPA1 synthesis in sensory neurons [15].

The role of reactive oxygen species in the pathogenesis of pruritic skin diseases such as atopic dermatitis and psoriasis is well elucidated in the current literature. These conditions are characterized by an imbalance of oxidant and antioxidant species, favoring oxidant species and consequent oxidative stress. Oxidant compounds such as hydrogen peroxide and tert-butylhydroperoxide (tBHP) can directly activate TRPA1 channels on dorsal root ganglia neurons. Furthermore, antioxidants have been shown to decrease scratching behavior evoked by these oxidant pruritogens, suggesting a possible role of the use of antioxidants in histamine-independent itch [6,13].

TRPA1 is thought to play a role in cholestatic itch. Intradermal injection of bile acid can induce itch and clearance of bile acids relieves it [6,20]. Itch signals are generated once bile acids bind to their receptor, TGR5, which is found in small-diameter dorsal root ganglion neurons that also express TRPA1. Lysophosphatidic acid (LPA) is a phospholipid that is found to be elevated in patients with cholestatic itch [16]. LPA utilizes TRPA1 to signal to neurons involved in itch [7]. LPA is synthesized by the enzyme autotaxin (ATX), and levels of ATX have been shown to positively correlate with itch severity in patients with cholestatic itch. Moreover, nasobiliary drainage decreases itch in patients with cholestatic itch, which is associated with a decrease in ATX following treatment [21]. Inhibiting the signaling pathway involving TRPA1 or TGR5 has been shown to improve or eliminate cholestatic itch in mice, and presents a potential target for future therapies [7].

TRPA1 can be activated by endogenous inflammatory mediators including bradykinin, arachidonic acid, and prostaglandins. The inflammatory leukotriene, leukotriene B4 (LTB4), is a potent activator of the TRPA1 channel [5]. LTB4 is released by white blood cells and acts as a potent chemoattractant to neutrophils and other inflammatory cells—it is found to be increased in atopic dermatitis and psoriasis, and contributes to itch by acting on specific GPCRs that then activate downstream TRPA1 channels [22].

Serotonin (5-HT) signaling plays a role in acute and chronic itch conditions through possible interactions with TRPA1. The serotonin receptor HTR7, can induce itch by activation from serotonin or by activation by the receptor agonist LP44, both of which signal to TRPA1 [6]. One study demonstrated that that by knocking out HTR7 or TRPA1, the itching effect of low-dose intradermal selective serotonin reuptake inhibition (to increase local serotonin) and LP44 was greatly attenuated. However, itch secondary to high-dose serotonin was not affected in HTR7 knockout mice, suggesting that this itch may be mediated by other players [12]. Another study demonstrated serotonin iontophoresis in skin inducing a non-histaminergic itch [6]. In mouse models with atopic dermatitis, knocking out TRPA1 and HTR7 reduced lesions and scratching in HTR7 and TRPA1 knockout mice with atopic dermatitis [7].

There are many other interesting findings regarding TRPA1 that may play a role in future therapies of chronic itch. TRPA1 has been shown to play a protective role in epidermal barrier recovery and homeostasis, with TRPA1 agonists found to promote skin barrier recovery [4,23]. Considering that many itchy diseases such as atopic dermatitis, contact dermatitis, and psoriasis involve epidermal barrier dysfunction, this could open the door for potential future therapies in these itch disorders. Periostin is an extracellular matrix protein that has been found to evoke itch in other animals such as monkeys, dogs, and mice, and is highly associated with chronic itch in humans suffering from atopic eczema, stasis dermatitis, and prurigo nodularis [16]. Periostin-induced itch is decreased by knocking out TRPV1 and TRPA1 in mice. Endothelin, a potent endogenous vasoconstrictor, can stimulate non-histaminergic itch receptors through activation of TRPA1. The endothelin-1 isoform, ET-1, was found to be increased in chronic itch. Knocking out TRPA1 in mice eliminated ET-1 mediated itch, and by blocking endothelin-converting enzyme 1, and subsequently increasing levels of ET-1, this resulted in increased itch [24]. Toll-like receptor 7 (TLR7) may play a role in histamine-independent itch and could be a target of immune therapies [6].

With regards to specific TRPA1 antagonists, there are few currently under study that can be potential treatments for itch [Table 2]. The TRPA1 antagonist, GRC 17536, finished phase II clinical trial and the initial results show that it is a promising treatment for diabetic neuropathy pain. This may also have possible use for nonhistaminergic itch, however, the antagonist did not advance into phase III clinical trials due to its pharmacokinetic profile and limited bioavailability [17]. Other potential novel treatments that antagonize the TRPA1 receptor include HC-030031, a TRPA1 antagonist that has been found to decrease itch in mouse models of atopic dermatitis, oxazolone and urushiol exposed mice, and LTB4-induced itch [16,17,23]. A-967079 is another TRPA1 antagonist that has been shown suppress itch in contact dermatitis mouse models. Additional study is required on the safety and efficacy of TRPA1 antagonists in treating chronic itch conditions [25].

## 3. TRP Vanilloid 1 (TRPV1)

TRPV1 is expressed in cutaneous and DRG sensory neurons involved in pain, thermoregulation, and pruritus [2]. It is also found in a variety of skin cells, where it plays a role in itch transmission, skin barrier homeostasis, epidermal proliferation, and inflammation [4]. These channels can be activated by endogenous molecules as well as acidic solutions, high temperatures, toxins, and chemicals such as capsaicin [2,16]. Capsaicin—one of the most well-studied TRPV1 activators— stimulates the release of substance P, a neuropeptide that causes an intense burning and stinging sensation that is perceived as either painful or itchy. Paradoxically, long-term use of capsaicin leads to analgesic and antipruritic effects due to desensitization of TRPV1-expressing afferent neurons [26]. The role of TRPV1 in itch seems to have both direct and indirect components. The TRPV1 channel plays a significant role in acute itch mediated by histamine [16,27]. Histamine directly induces itch through TRPV-1-expressing C-fiber nociceptors via histamine GPCRs (H1R and H4R), which activate the phospholipase C beta-3 pathway [16]. Inhibition of TRPV1 has been shown to reduce H1R- and H4R-induced itch [27]. Furthermore, endogenous pruritogens such as ATP, lipoxygenase products, and prostaglandins are known to potentiate TRPV1 activity of sensory afferents [4]. TRPV1 also plays an indirect role in non-histaminergic pruritus, which is the main pathway of chronic itch. Proteases activate their respective protease activated receptors (PARs), such as PAR2 and PAR4, which are involved in chronic neurogenic inflammation. This inflammation sensitizes TRPV1 channels and induces itch [16]. Itch signaling mediated by interleukins in neurons has also been shown to require TRPV1. In one study, IL-31-induced pruritus was significantly decreased in TRPV-1 deficient mice [28]. Liu et al. found that administering TRPV1 blockers drastically reduced IL-33 pruritic responses in dorsal root ganglia neurons [29]. These interleukins are known to be associated with pruritic disorders such as atopic dermatitis and allergic contact dermatitis, thus highlighting the role of TRPV1 channels in these conditions. Increased expression and phosphorylation of TRPV1 has also been observed in atopic dermatitis lesions [30]. TRPV1 may also play a role in psoriatic itch. Imiquimod, which has been previously used for generating psoriasis murine models, requires TRPV1-expressing neurons to cause pruritus [16]. Furthermore, TRPV1 gene overexpression was found to be positively correlated to itch intensity in patients with atopic dermatitis and psoriasis [11]. TRPV1 expression is also increased in keratinocytes and sensory nerves of pruritic prurigo nodularis lesions [4].

Modulation of TRPV1 channels as a treatment for pruritic diseases has been extensively studied. Topical capsaicin is often prescribed to treat neuropathic pruritus associated with notalgia paresthetica [31]. However, the burning and discomfort associated with application can be very bothersome, and symptoms have been shown to relapse with discontinuation of treatment in some patients [31,32]. While oral TRPV1 antagonists showed promise in preclinical trials, they caused significant hyperthermia in patients in phase I randomized controlled trials and clinical development of these systemic agents was discontinued. This adverse effect was thought to be due to TRPV1′s role in thermoregulation [33]. Administration of a topical TRPV1 antagonist markedly reduced scratching behavior, as well as erythema and edema, in atopic dermatitis mice models by day 12 of treatment, a finding previously supported by other murine studies [34,35]. Topical TRPV1 antagonist PAC-14028 has been assessed in phase II randomized controlled trials for the treatment of mild to moderate atopic dermatitis, showing significant reduction in pruritus-related VAS scores by week 8 [36]. A phase III randomized controlled trial recently evaluated the use of Asivatrep, a topical TRPV1 antagonist, for the treatment of atopic dermatitis. Patient-reported pruritus assessments and pruritus VAS scores at week 8 were lower in patients treated with Asivatrep compared to the vehicle. Hyperthermia was not reported with use of this topical treatment, suggesting its favorable safety profile [37].

## 4. TRP Vanilloid 2 (TRPV2)

Similar to TRPV1 channels, TRPV2 channels are expressed on sensory neurons that are predominately found in between the papillary dermis and the epidermis around hair follicles, and blood vessels in the dermis; they are also found in non-neuron type cells of the skin such as in keratinocytes and macrophages [4,5]. These channels are activated at temperatures greater than or equal to 52 degrees Celsius in sensory neurons. TRPV2 is also expressed on mast cells, and increased temperature or physical stimuli can activate TRPV2, resulting in degranulation that leads to an inflammatory cascade involving downstream activation of protein kinase A signaling—one of the most important mechanisms for initiating sensitization of pain and itch receptors [5]. Mast cell degranulation secondary to TRPV2 channel activation was found to be inhibited by the nonselective TRPV2 antagonist, SKF96365 [38]. Targeting TRPV2-mediated mast cell degranulation presents as a possible target for itch, although further studies are needed. 

## 5. TRP Vanilloid 3 (TRPV3)

TRPV3 is primarily expressed in keratinocytes in the skin, but it can also be found in nasal and oral epithelium. A variety of ligands activate this channel, including heat, exogenous plant-derived compounds, and endogenous molecules such as arachidonic acid and farnesyl pyrophosphate (FPP) [2,7]. In the skin, TRPV3 plays a role in both pain and pruritus pathways. Missense, gain-of-function TRPV3 mutations have been discovered in Olmsted syndrome, a genetic condition associated with skin barrier defects and keratoderma that is extremely itchy. In mice, this mutation causes increased scratching behavior, further indicating that TRPV3 may be involved in this syndrome’s pruritogenic pathway [7]. Inhibition of TRPV3 has been shown to attenuate atopic itch, and patients with atopic dermatitis who experienced pruritus were found to have higher levels of TRPV3 mRNA expression [16,39]. The mechanism of itch involving TRPV3 may be associated with PAR2 receptors, for it has been shown that keratinocytes lacking TRPV3 impair PAR2 function, resulting in decreased neuronal activation and scratching behavior in response to PAR2 agonists [39]. Another potential pruritogenic mechanism involves IL-31-mediated B-type natriuretic peptide (BNP) synthesis, a molecule which has been shown to induce atopic dermatitis skin inflammation. It was shown that BNP binds to keratinocytes and upregulates TRPV3 gene transcription, resulting in downstream signaling that promoted itch transduction [40]. TRPV3 channels may also play a role in psoriatic itch. While one study found increased TRPV3 expression in both lesional atopic dermatitis and lesional psoriasis skin when compared to healthy controls, another study examining the itch transcriptome of eczema and psoriasis found increased TRPV3 gene expression in pruritic psoriatic skin only [11,40]. Naturally occurring plant compounds such as coumarin osthole, verbascoside, and citrusinine-II have been shown to selectively inhibit TRPV3 and significantly attenuate pruritus [41,42,43]. TRPV3 antagonist dyclonine improves skin inflammation and abrasions, pain, and pruritus, and is FDA-approved as a topical anesthetic [44]. Trpvicin, a selective TRPV3 antagonist, has been shown to reduce scratching behavior in mice models of acute and chronic pruritus [45]. Furthermore, a recent novel topical small molecular inhibitor of TRPV3 channel (KM001) is currently undergoing phase II trials for lichen simplex chronicus, which is highly associated with keratinocyte itch activation [46].

## 6. TRP Vanilloid 4 (TRPV4)

TRPV4 is activated by physical stimuli such as osmotic changes and heat, as well as endogenous and synthetic ligands [2]. It is expressed in both dorsal root ganglia sensory neurons and skin cells, where it plays a role in the pathophysiology of pruritus. TRPV4 in epidermal keratinocytes has been shown to mediate acute histaminergic itch as well as itch evoked by endothelin-1, but its role in chloroquine-induced nonhistaminergic pruritus remains a point of contention [7,47]. Similar findings were elicited when examining TRPV4 channels in sensory neurons [48]. Direct activation of this channel with a selective agonist evoked scratching behavior that was dependent on TRPV4 expression in keratinocytes, further elucidating its role in the itch pathway [47]. Serotonin (5-HT)-induced pruritus has also been shown to involve TRPV4 signaling. TRPV4-deficient mice were shown to have significantly fewer 5-HT-induced scratching episodes when compared to controls, and pretreatment with a TRPV4 antagonist was found to decrease pruritus induced by serotonin in vivo [7]. These findings suggest that TRPV4 antagonists could function as anti-pruritic treatments in itchy conditions involving 5-HT such as atopic dermatitis, psoriasis, and contact dermatitis [16]. Recent studies suggest that TRPV4 may play a role in these dermatologic conditions as well as in other forms of chronic pruritus. In mice models of chronic itch, dry skin was shown to increase TRPV4 expression, and scratching was significantly reduced in mice lacking TRPV4. Furthermore, dry skin-induced pruritus was attenuated by TRPV4 selective antagonists [47]. TRPV4 expression in skin cells was upregulated in allergic contact dermatitis models [49]. Expression was also upregulated in both the epidermis and dorsal root ganglia neurons in psoriasis models [49,50]. In humans, TRPV4 mRNA was found to be increased in skin biopsies of patients with chronic idiopathic pruritus when compared to controls [49]. This channel is also involved in chronic pruritus due to systemic causes. Lysophosphatidylcholine (LPC)—the precursor for LPA—is a cholestatic pruritogen. This molecule induces itch by directly activating TRPV4 in keratinocytes, which then release a microRNA that activates TRPV1 in sensory neurons [51]. Given that TRPV4 is expressed widely throughout the body and functions in a variety of biological processes, selective inhibition of itch related TRPV4 channels would be a potential treatment of pruritic disorders. 

## 7. TRP Melastatin 8 (TRPM8)

TRPM8 can be found in a variety of organs and tissues, including peripheral sensory neurons where it plays a role in non-noxious cold sensation. It is also activated by chemical compounds known to produce cooling sensations, such as menthol and icilin [52]. However, in contrast with other TRP channels, TRPM8 activity suppresses itch rather than inducing it in the majority of the cases. Topical cooling has been used to reduce pruritus, for cooling decreases nerve excitability and conduction velocity and therefore slows some itch transduction pathways, such as the one involving TRPV1 [53]. Cold temperatures and menthol, however, also excite sensory neurons that express TRPM8 [53]. Palkar et al. showed that cooling successfully inhibits both histaminergic and non-histaminergic itch pathways, and that this mechanism requires activation of TRPM8 channels or TRPM8-expressing afferent neurons [54]. Although they are not pruriceptors, TRPM8 neurons are thought to play a role in suppressing itch by participating in a spinal interneuron circuit involving B5-I neurons. These inhibitory spinal interneurons receive input from menthol-sensitive afferents and produce dynorphin, a neuropeptide known to suppress itch [54]. It has been shown that menthol was unable to inhibit pruritus in mice lacking B5-I neurons, suggesting that these neurons play a role in the antipruritic effects involving TRPM8 activation [55].

Cooling has been shown to inhibit chronic itch, thus making TRPM8 a potential target for antipruritic therapy [54]. In a double-blind, randomized study, a topical cooling lotion composed of two TRPM8 agonists was evaluated for treatment of chronic pruritus in patients with dry skin. Up to 84% of patients using the lotion reported greater improvement of pruritus compared to those using the vehicle [56]. Cryosim-1, a topical synthetic TRPM8 agonist, has been evaluated in randomized controlled trials for the treatment of pruritus. One study showed significant decrease in pruritus NRS scores two hours after Cryosim-1 application in patients with atopic dermatitis and urticaria [57]. In another randomized controlled trial evaluating Cryosim-1 gel for scalp pruritus, pruritus NRS scores significantly decreased two hours after cryosim-1 application when compared to the vehicle, and this rapid itch relief translated into an improved quality of life in those patients [58]. Interestingly, TRPM8 expression was found to be increased in patients with pruritic psoriatic skin and correlated to itch intensity in both plaque and scalp psoriasis [11]. Therefore, further studies should be conducted to evaluate topical TRPM8 agonists in the treatment of certain types of pruritus, as it could aggravate itch in some conditions such as psoriasis.

## 8. TRP Cannonical 3 and 4 (TRPC3, TRPC4)

TRPC4 is an important mediator of keratinocyte differentiation and is found in keratinocytes [4]. Selective serotonin reuptake inhibitors (SSRIs) have been reported to cause many different dermatologic side effects including itch although in humans they are used successfully as anti pruritics. Subcutaneous injection of sertraline, a common SSRI medication, elicited a strong itch response in mouse models [12,17]. TRPC4 has been shown to play a role in sertraline-induced itch as evidenced by decreased itch after genetically targeting TRPC4. The serotonin receptor HTR2B is also associated with this pathway as illustrated by targeting this receptor causing reductions in sertraline-induced itch when compared to control mice. There are a few small molecule TRPC4 antagonists available for study including ML204, M084, and HC-070, however, they have been shown to act in a nonselective manner and inhibit other TRPC channels [17,59]. TRPC3 is another channel that may play a role in many different forms of itch including chloroquine-induced itch, beta-alanine itch, and itch associated with contact dermatitis [60]. One study found that TRPC3 agonism induced histamine-independent itch and TRPC3 knockout mice had significantly decreased spontaneous scratching behaviors. Additional research is needed on the therapeutic benefits of targeting TRPC channels [61].

## 9. Possible Ethnic Differences in TRP Channels and Itch

We have demonstrated that there are significant differences in response to capsaicin, a TRPV1 agonist, among different ethnic groups. Hispanics reported significant itch while African Americans demonstrated a significantly decreased sensitivity to heat pain [62]. We have also observed that there is a high sensitivity to cold stimuli in African Americans. A large study was conducted on TRPA1 gene polymorphism and its genetic contribution to pain [63]. These results suggest that there may be genetic polymorphism of TRPV1, TRPA1, and TRMP8 in ethnic populations that may affect therapy response to anti-pruritic topical treatments targeting TRP channels. Further controlled studies on larger cohorts will be required to study possible genetic polymorphism in TRP channels in ethnic populations.

## 10. Conclusions

As TRP channels integrate several mechanisms of itch and inflammation, and are abundant in the skin, they may serve as potentially excellent targets for topical anti-pruritic drug treatments. In particular, drugs that target TRPV1, TRPA1, and TRPV3, as well as cooling agents that target TRPM8 with more long-lasting effects and less irritation than menthol, are all of great interest.

## Figures and Tables

**Figure 1 ijms-24-00420-f001:**
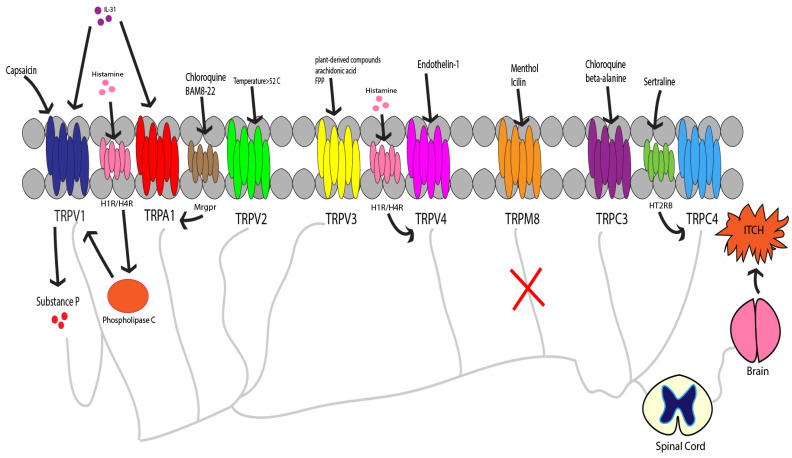
Overview of itch related TRP channels.

**Table 1 ijms-24-00420-t001:** TRP channels involved in itch and their associated pathways and diseases.

TRP Channel	Activating Compounds	Associated Pathways/Diseases
**TRPA1**	Chloroquine, cowhage, allyl isothiocyanate, cinnamaldehyde, allicin, carvacrol, LTB4, bradykinin, arachidonic acid, prostaglandins, TSLP, 5-HT, bile acids, LPA, IL-31, IL-13, BAM8-22, hydrogen peroxide, tBHP, endothelin	Mrgpr-associated nonhistaminergic pruritus, PAR-mediated nonhistaminergic pruritus; AD, allergic contact dermatitis, cholestatic pruritus, psoriasis
**TRPV1**	Capsaicin, histamine, ATP, lipoxygenase products, prostaglandins, imiquimod	Histaminergic pruritus, PAR-mediated nonhistaminergic pruritus, IL-31, and IL-33 mediated itch pathways; AD, psoriasis, prurigo nodularis
**TRPV2**	Increased temperature, physical stimuli	Mast cell degranulation, PKA-mediated inflammatory cascade
**TRPV3**	Plant-derived compounds, arachidonic acid, farnesyl pyrophosphate	PAR-mediated nonhistaminergic pruritus, IL-31-mediated BNP synthesis; Olmsted syndrome, AD, psoriasis
**TRPV4**	Histamine, endothelin-1, 5-HT, Lysophosphatidylcholine (LPC)	Histaminergic pruritus; dry skin pruritus, allergic contact dermatitis, psoriasis, chronic idiopathic pruritus
**TRPM8**	Menthol, icilin	Histaminergic and nonhistaminergic pruritus, B5-I neuron-associated spinal interneuron circuit; dry skin pruritus, AD, urticaria, scalp pruritus
**TRPC3**	Chloroquine, beta-alanine	Nonhistaminergic pruritus; Contact dermatitis
**TRPC4**	Sertraline	Serotonin receptor HTR2B-associated itch

**Table 2 ijms-24-00420-t002:** TRP channel agonists and antagonists.

TRP Channel	Antagonists	Notes
**TRPA1**	GRC 17536, HC-030031, A-967079	Efficacy tested in diabetic neuropathy models, AD models, contact dermatitis models, LTB4-induced itch
**TRPV1**	Asivatrep, PAC-14028	Efficacy tested in AD models, AD patients
**TRPV2**	SKF96365	Inhibits mast cell degranulation secondary to channel activation
**TRPV3**	coumarin osthole, verbascoside, citrusinine-II, dyclonine, trpvicin, KM001 *	KM001 undergoing trial for lichen simplex chronicus
**TRPV4**	HC067047	Efficacy tested in dry skin pruritus models
**TRPC3**	None reported	
**TRPC4**	ML204, M084, HC-070	Act in non-selective manner
**TRP Channel**	**Antagonists**	**Notes**
**TRPM8**	(1R,2S,5R)-N-(2-(2 pyridinyl) ethyl)-2-ispropyl-5-methylcyclohexancarboxamide, menthoxypropanediol, Cryosim-1, Icilin	Efficacy tested in dry skin pruritus, AD, urticaria, scalp pruritus

* Undergoing clinical trials.

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
