# Peer review of "Transient Receptor Potential Channels and Itch"

_ijms, 2022, doi:10.3390/ijms24010420_

Round 1
Reviewer 1 Report
Omar et al. summarized the TRP members’ roles, and their antagonists involved in itch signaling, And they highlighted the promising targets advancing into the clinical trials and discussed the ethnic differences in TRP channel genetic polymorphisms which may affect the treatment response. This manuscript was well-written and prepared. I would recommend it as a minor revision.
Here are my suggestions before publication:
1. Add a few sentences more in the background of Itch itself and TRP channels in the Introduction part.
2. If possible, add a figure drawing brief pathways of TRP channels and itch signaling, which can let readers have a quick view.
3. As well as A table of the antagonists/targets is needed. Highlight the ones in the trials.
Reviewer 2 Report
The paper concerns topics that were previously described in other papers. Other reviews covers the topic (e.g. Moore C, Gupta R, Jordt S-E, Chen Y, Liedtke WB. Regulation of pain and itch by TRP channels. Neuroscience bulletin. 2018;34(1):120-142; Xie Z, Hu H. TRP channels as drug targets to relieve itch. Pharmaceuticals. 2018;11(4):100;
In my opinion the paper does not extend the existing knowledge and is not a significant contribution to the field.
The paper seems to be disorganized. TRPV receptors should be next to each other in the text. The authors should consider including some tables or graphs which allow to organize information presented in the paper.
Line 154 – allicin with a lowercase letter
Round 2
Reviewer 2 Report
I appreciate the changes and - what is especially important – the new tables and figure which improved this manuscript (Table 2 caption should be more informative eg. TRP channel agonists and antagonists and its efficacy as drugs). The main text was not changed substantially, and I’m not sure: what is the purpose of writing this paper other than repeating the already existing knowledge? Which information here is new? The answer from the authors that ‘This paper is intended to be a review article’ does not convince me. Therefore, I leave the final decision whether publish this paper to the Editor.